# Knowledge, attitude, perception, and preventative practices towards COVID-19 in sub-Saharan Africa: A scoping review

Ugochinyere Ijeoma Nwagbara[1]*, Emmanuella Chinonso Osual[2], Rumbidzai Chireshe[1], Obasanjo Afolabi Bolarinwa[1,3], Balsam Qubais Saeed[4], Nelisiwe Khuzwayo[5], Khumbulani W. Hlongwana[1]

1 Discipline of Public Health Medicine, School of Nursing and Public Health, College of Health Sciences, University of KwaZulu-Natal, Howard Campus, Durban, South Africa, 2 Discipline of Pharmaceutical Sciences, School of Nursing and Public Health, College of Health Sciences, University of KwaZulu-Natal, Westville Campus, Durban, South Africa, 3 Department of demography and social statistics, Faculty of Social Sciences, Obafemi Awolowo University, Nigeria, 4 Department of Clinical Sciences, College of Medicine, University of Sharjah, Sharjah, UAE, 5 Discipline of Rural Health, School of Nursing and Public Health, College of Health Sciences, University of KwaZulu-Natal, Howard Campus, Durban, South Africa

* 216045259@stu.ukzn.ac.za, ugochinyereijeoma@gmail.com

**Data Availability Statement:** All relevant data are within the manuscript and its Supporting Information files.

## Abstract

### Background

Knowledge, attitudes, perception, and preventative practices regarding coronavirus- 2019 (COVID-19) are crucial in its prevention and control. Several studies have noted that the majority of people in sub-Saharan African are noncompliant with proposed health and safety measures recommended by the World Health Organization (WHO) and respective country health departments. In most sub-Saharan African countries, noncompliance is attributable to ignorance and misinformation, thereby raising questions about people's knowledge, attitudes, perception, and practices towards COVID-19 in these settings. This situation is particularly of concern for governments and public health experts. Thus, this scoping review is aimed at mapping evidence on the knowledge, attitudes, perceptions, and preventive practices (KAP) towards COVID-19 in sub-Saharan Africa (SSA).

### Methods

Systematic searches of relevant articles were performed using databases such as the EBSCOhost, PubMed, Science Direct, Google Scholar, the WHO library and grey literature. Arksey and O'Malley's framework guided the study. The risk of bias for included primary studies was assessed using the Mixed Method Appraisal Tool (MMAT). NVIVO version 10 was used to analyse the data and a thematic content analysis was used to present the review's narrative account.

### Results

A total of 3037 eligible studies were identified after the database search. Only 28 studies met the inclusion criteria after full article screening and were included for data extraction.

**Funding:** The author(s) received no specific funding for this work.

**Competing interests:** The authors have declared that no competing interests exist.

**Abbreviations:** COVID, Coronavirus 2019; KAP, Knowledge, attitude, perception and practice; MESH, Medical Subject Headings; MMAT, Mixed Method Appraisal Tool; SSA, sub-Saharan Africa; WHO, World Health Organization.

Studies included populations from the following SSA countries: Ethiopia, Nigeria, Cameroon, Uganda, Rwanda, Ghana, Democratic Republic of Congo, Sudan, and Sierra Leone. All the included studies showed evidence of knowledge related to COVID-19. Eleven studies showed that participants had a positive attitude towards COVID-19, and fifteen studies showed that participants had good practices towards COVID-19.

## Conclusions

Most of the participants had adequate knowledge related to COVID-19. Despite adequate knowledge, the attitude was not always positive, thereby necessitating further education to convey the importance of forming a positive attitude and continuous preventive practice towards reducing contraction and transmission of COVID-19.

## Introduction

The coronavirus-2019 (COVID-19), which is also referred to as serious acute respiratory syndrome coronavirus-2 (SARS-CoV-2), was first reported in Wuhan, China, in December 2019 [1, 2]. The World Health Organization (WHO) declared COVID-19 as a global pandemic on March 11, 2020, due to its continuous global spread [3]. COVID-19 is considered a zoonotic infectious disease that can spread amongst humans or animals to human [1]; when transmitted by humans, it could lead to serious respiratory conditions [3, 4]. The key clinical signs and symptoms include fatigue, a fever of 39 degrees and above, dry cough, dyspnoea, fatigue, and myalgia, and in some severe cases, COVID-19 infection can cause kidney failure, severe pneumonia and acute respiratory syndrome, and even death [3]. COVID-19 confirmed global cases as of November 15, 2020 was 53,976,457, with 1,311,942 deaths and 34,772,744 recoveries [5].

Several studies have demonstrated that the main mode of transmission of COVID-19 is through respiratory droplets of an infected person when they sneeze or cough [6–9]. Even though the spread of COVID-19 is at its peak in most European and American countries, it is still accelerating in most African countries [9, 10]. The high infection rate in sub-Saharan Africa can present a much difficult situation because of different comorbidities combined with poverty, poor healthcare services and limited access to health facilities [9, 11]. To control and prevent contracting and spreading COVID-19, people need to possess appropriate knowledge regarding the disease, have correct attitude and follow correct practices against the virus. A study conducted in Jimma town, Ethiopia in 2020, showed that a larger percentage of the participants knew the key clinical symptoms and mode of transmission of COVID-19 and that older people who have chronic illnesses were at high risk of developing a severe form of coronavirus-19 [12]. Another study in Africa noted a huge disparity in knowledge, attitude and practice towards the virus [13]. Further studies conducted in SSA reported that residents were noncompliant with proposed health and safety measures recommended by the WHO and respective country health departments. This was due to ignorance and misinformation [9, 10]. A scoping review of literature on KAP studies conducted in SSA towards COVID-19 is critical in order to develop tailored interventions. Therefore, it is anticipated that the results of this study will reveal research gaps to guide health experts in decision-making in SSA, as well as develop policies and interventions tailored towards bridging the gap.

## Materials and methods

The study adopted a scoping review because of its ability in mapping of new concepts, types of evidence and associated gaps on available evidence [14]. The Arksey and O'Malley's methodological framework for scoping reviews involving: (i) identifying the research question, (ii) identifying relevant studies, (iii) study selection, (iv) charting the data, and (v) collating, summarising and reporting results [15], guided this review. The review protocol for this scoping review has been registered on the Open Science Framework (OSF) and can be accessed via https://osf.io/sdm46/. To ensure the quality of the primary studies included in this study, we utilised the quality assessment recommended by Levac et al. [16]. The PRISMA (Preferred Reporting Items for Systematic and Meta-Analysis) extension for scoping reviews (PRISMA-ScR) was used [17] as presented in **S1 Appendix**.

### 1. Identifying the research question

The research question was, "what is known from existing literature on the knowledge, attitudes, perceptions and preventative practice towards COVID-19 in SSA?"

### 2. Identifying relevant studies

This scoping review covered all studies published in peer-reviewed journals and grey literature addressing the above-stated research question. Systematic searches of relevant articles were performed using electronic databases such as PubMed, Google Scholar, Science Direct, EBSCOhost platform and the World Health Organization (WHO) library. Identification of studies was accomplished by searching published literature in the English language between December 2019 and October 2020. These timelines were motivated by the onset of the COVID-19 pandemic and the timing for initiating this review. The search terms included were 'Knowledge', 'Attitude', 'Perception', 'Practice', 'Coronavirus 2019', and 'sub-Saharan Africa'. Boolean terms such as 'OR' and 'AND' were used to separate the search keywords. Medical Subject Headings (MeSH) terms were also included in the search (see "**S2 Appendix**" for further details).

### 3. Study selection

Following a comprehensive title screening from the databases mentioned above, all studies that did not address the study's research question were excluded along with all the duplicates. All included studies for abstract screening were uploaded on Endnote X9 software. The inclusion and exclusion criteria formulated according to the research questions were used to identify the relevant studies. Two independent reviewers (OAB and ECO) screened the abstracts and full articles, and a third screener (UIN) resolved all the discrepancies between reviewers at the abstracts and full articles screening stages before including the study in the review.

**Inclusion criteria.** The inclusion criteria include:

- Original research articles reporting information regarding knowledge, attitude, perceptions and practices towards COVID-19.

- Articles published in peer-reviewed journals and grey literature with any study design addressing the research question.

- Articles published in English.

- Articles published between December 2019 and October 2020.

- Articles involving participants recruited from sub-Saharan Africa countries.

**Exclusion criteria.** The exclusion criteria are as follows:

- Any studies that did not report on the knowledge, attitudes, perceptions and practices towards COVID-19.

- Any studies conducted on COVID-19, but among infants, given the mild nature of the disease in infants.

- Studies not conducted in sub-Saharan Africa countries.

- Any studies published in languages other than the English language were excluded.

## 4. Charting the data

We used NVivo version 10 to organize data extracted from included studies into different themes [18]. We extracted data on the following headings: author and year, study setting (country), study design, population, mean/age range of participants, percentage of males, percentage of females, knowledge related to COVID-19, attitude/perception towards COVID-19, practice towards COVID-19 and Relevant findings as shown on **Table 1**.

## 5. Collating, summarising and reporting the results

Thematic content analysis was used to analyse the narrative account of the data extracted from the included studies. Data was extracted around the following outcomes: knowledge of COVID-19, attitude towards COVID-19, perceptions towards COVID-19 and preventative practices towards COVID-19.

**Quality appraisal.** We performed the quality assessment of included studies using the Mixed Methods Appraisal Tool (MMAT) Version 2011 to assess the risk of bias for the included primary studies [19]. Two independent reviewers (OAB and UIN) assessed the quality of the included studies, using the following domains: the appropriateness of the research question, data collection, data analysis, accuracy of the sampling methodology, author's acknowledgement of possible biases and conclusion. An overall percentage quality score for each of the included studies was calculated and interpreted as <50% (low quality), 51–75% (average quality) and 76–100% (high quality).

## Results

### Screening results

A total of 3037 eligible studies were identified from the databases searched (**Fig 1**). One hundred and sixty-four (164) studies were retained after duplicate removal and title screening. A total of 128 studies were excluded after the abstract screening conducted by two researchers, thus reducing the articles eligible for full-article screening to 36 articles. After the full-article screening, 8 studies were excluded, of which 5 studies did not report on our intervention of interest which was KAP regarding COVID-19 [20–24], 2 studies were conducted in SSA, but included countries that are not in SSA [25, 26], and 1 study was a presentation and not an article [27]. In total, 28 articles were finally included for data extraction in the review.

The Preferred Report Items for Systematic and Meta-Analysis (PRISMA) flow chart for the screening and selection of studies in this review is shown in **Fig 1**.

**Table 1.** Summary characteristics of the included studies.

| Author and year | Study setting (Country) | Study design | Population (n) | Mean/Age range of participants | Percentage (%) of males | Percentage (%) of females | Knowledge related to COVID-19 | Attitude/Perception towards COVID-19 | Practice towards COVID-19 | Relevant findings |
|---|---|---|---|---|---|---|---|---|---|---|
| Adela et al 2020 [45] | Cameroon | Cross-sectional survey | 1006 participants | 33 | 46.9% | 53.1% | The participants had a high overall knowledge score of 84.19% | The overall score was 69% for attitude | The overall score was 60.8% for practice towards COVID-19 | There was high knowledge and perception of COVID-19 disease transmission |
| Adhena and Hidru and 2020 [29] | Ethiopia | Cross-sectional study | 419 participants | 69.5 | 51.3% | 48.7% | About 37.7% of participants had poor knowledge | About 43.4% of participants had a negative attitude towards COVID-19 | About 52.5% of the participants had poor practice towards COVID-19 | The overall KAP regarding COVID-19 prevention and control was shown to be poor |
| Akalu et al 2020 [10] | Ethiopia | Cross-sectional study | 404 participants | 56.5 | 60.9% | 39.1% | Poor knowledge was reported in 33.9% of the participants | About 36.1% of the study participants were of the perception that they have a moderate risk of COVID-19 infection | 47.3% of the chronic disease patients had poor practices | Most participants revealed poor knowledge and poor practice |
| Anikwe et al 2020 [37] | Nigeria | Cross-sectional survey | 430 participants | 30.04 | N/A | 100% | Most of the women showed adequate knowledge about COVID-19 infection | Majority of participants showed a good attitude towards COVID-19 | Majority of the women had good preventive practice towards COVID-19 disease | The study population (pregnant women) had good KAP towards COVID-19 disease |
| Asemahagn and 2020 [30] | Ethiopia | Cross-sectional survey | 398 participants | 34 | 58.0% | 42.0% | This study reported that 70% of the Healthcare workers had good COVID-19 related knowledge | N/A | Study findings revealed that 62% of the Healthcare workers had good COVID-19 preventive practice | Majority of the Healthcare workers reported good knowledge of COVID-19 but had lower preventive practice |
| Asmelash et al and 2020 [31] | Ethiopia | Cross-sectional study | 410 participants | 47 | 92.4% | 7.6% | Of the total participants, 60.7% had good knowledge | Of the total participants, 34.1% showed a positive attitude towards COVID-19 | Few of the study participants (15.6%) had good preventive practices towards COVID-19. | Most of the participants had negative attitudes and poor practice towards COVID-19. |
| Girma et al 2020 [32] | Ethiopia | A Web-Based Survey | 273 participants | 31.03 | 89% | 11% | All participants in the study correctly answered all preventive knowledge questions | N/A | Participants had low mean scores for precautionary behavior questions such as wearing a mask and wearing gloves and the highest score for avoiding people who are sneezing or coughing | There was a substantial gap in the knowledge level and execution of behavioral practice, particularly wearing masks and gloves. |

*(Continued)*

**Table 1.** (Continued)

| Author and year | Study setting (Country) | Study design | Population (n) | Mean/Age range of participants | Percentage (%) of males | Percentage (%) of females | Knowledge related to COVID-19 | Attitude/Perception towards COVID-19 | Practice towards COVID-19 | Relevant findings |
|---|---|---|---|---|---|---|---|---|---|---|
| Haftom et al and 2020 [33] | Ethiopia | Cross-sectional study | 331 participants | 30.5 | 69.5% | 30.5% | Below half (42.9%) of the participants were knowledgeable about COVID-19 | About one-third of the participants responded that the Ethiopian government is handling the COVID-19 pandemic crisis well | Low report of any practices related to COVID-19 | A significant number of participants lacked knowledge and poorly adhered to COVID-19 prevention strategies |
| Iradukunda and 2020 [49] | Rwanda | Cross-sectional study | 376 participants | 38 | 44% | 56% | A high percentage of the participants, (n = 363, 97%) got a high knowledge score | Over one-quarter of the study participants (26%) had a poor attitude score | Most participants (90%) had a high practice score | The study findings showed a high knowledge and practice score towards COVID-19 and a poor attitude score |
| Kassie et al and 2020 [34] | Ethiopia | Cross-sectional study | 408 participants | 30.33 | 67.3% | 32.7% | The participants with good Knowledge constituted 73.8% | Almost two-thirds (65.7%) of the healthcare providers showed positive attitude towards COVID-19 | N/A | The health care providers had good COVID-19 related knowledge and attitude. |
| Kebede et al and 2020 [12] | Ethiopia | Cross-sectional study | 247 participants | 30.5 | 76.5% | 23.5% | Overall, the proportion of visitors with high knowledge was 41.3% | Majority of the visitors felt self-efficacious in controlling COVID-19 (68.8%) while 83.3% believed that COVID-19 is a stigmatized disease | The main practices observed by the visitors were frequent washing of hands (77.3%) and avoidance of shaking hands (53.8%). | The visitors had moderate knowledge, perceived self-efficacy in controlling COVID-19 and preventive practices against the contagious virus |
| Carsi et al and 2020 [51] | Democratic Republic of Congo | Cross-sectional study | 347 participants | 37.4 | 17% | 83% | Less than one-third of the respondents (30%) had correct COVID-19 knowledge | This survey indicates that most of the participants (88%) did not agree that the COVID-19 situation would be under control and defeated in the DRC | Most participants did not engage in handwashing, wearing of facemasks and social distancing | Preventive practices were seldom in place |

*(Continued)*

**Table 1.** (Continued)

| Author and year | Study setting (Country) | Study design | Population (n) | Mean/ Age range of participants | Percentage (%) of males | Percentage (%) of females | Knowledge related to COVID-19 | Attitude/ Perception towards COVID-19 | Practice towards COVID-19 | Relevant findings |
|---|---|---|---|---|---|---|---|---|---|---|
| Mandaah et al and 2020 [46] | Cameroon | Cross-sectional study | A total of 480 and 680 participants were sampled at onset and two months later respectively | 20–29 years | Onset-23.2% Two months after- 32.0% | Onset- 18.2% Two months after- 26.6% | The overall proportion of people with correct knowledge moved from 9.1% at onset to 41.4% after two months | Overall, there was a positive change in the attitude of the people towards COVID-19 as the disease progressed | The participants' practices with regards to COVID-19 showed some improvement two months after the pandemic started | There was positive change in the KAP of the population regarding COVID-19 two months after the start of the pandemic |
| Mbachu et al and 2020 [38] | Nigeria | Cross-sectional study | 403 participants | 36.69 | 45.7% | 54.3% | Three hundred and fifty-seven (88.59%) of the participants had good knowledge of COVID-19 | A substantial proportion of the healthcare workers had either poor (n = 101, 25.06%) or indifferent attitude to work (n = 233, 57.82%) in the COVID-19 era | Three hundred and twenty-eight HCWs (81.39%) had a high level of practice towards preventing COVID-19 infection | Good knowledge which influenced practice and high level of practice of preventive measures, with associated poor attitude was observed among the healthcare workers |
| Mousa et al and 2020 [52] | Sudan | Cross-sectional study | 2336 participants | 18 and 29 years | 39.3% | 60.7% | The participants' mean knowledge score was 84.7%. | Most of the participants (94.8%) were willing to commit to staying at home | A large percentage of the participants (92%) frequently washed their hands or used antiseptic | Participants who were young, and especially females, had good knowledge, hopeful attitudes, and acceptable practices towards COVID-19 |
| Nicholas et al and 2020 [47] | Cameroon | Cross-sectional study | 480 participants | 18 years and above | 56.0% | 44.0% | Of the 545 study participants, 21.9% had a correct COVID-19 knowledge | The population generally had a good attitude towards COVID-19 disease | At least one preventive measure of the disease was known to all the participants, but the number of preventive measures known differed from one participant to another | There is a gap in the knowledge about COVID-19 among the Buea population |
| Nkansah et al and 2020 [50] | Ghana | Cross-sectional study | 261 participants | 32.0 | 50.6% | 49.4% | About two-thirds of the healthcare workers (65.1%) had adequate knowledge about COVID-19 | N/A | Generally, 57.5% of the participants practiced precautionary behaviour adequately | The healthcare workers had encouraging knowledge, practice and willingness to handle COVID-19 |

(*Continued*)

**Table 1.** (Continued)

| Author and year | Study setting (Country) | Study design | Population (n) | Mean/ Age range of participants | Percentage (% ) of males | Percentage (%) of females | Knowledge related to COVID-19 | Attitude/ Perception towards COVID-19 | Practice towards COVID-19 | Relevant findings |
|---|---|---|---|---|---|---|---|---|---|---|
| Nwonwu et al and 2020 [39] | Nigeria | Cross-sectional study | 320 participants | 41.6 | 52.5% | 47.5% | Majority of the respondents (n = 256, 80.0%) had good COVID-19 knowledge | N/A | Only 133 (41.6%) of the respondents had good preventive practices | Despite the good knowledge of COVID-19 by the respondents, preventive measures against the disease were generally poor |
| Ogolodom et al and 2020 [40] | Nigeria | Descriptive study | 300 participants | 33.6 | 42.7% | 57.3% | Most of the healthcare workers, 168 (56%) were highly aware of the pandemic | Most of the participants 183 (61%) felt they were at risk of being infected by the virus. | N/A | The health care workers are highly aware of the etiology, mode of transmission and symptoms of coronavirus disease |
| Ojo et al and 2020 [41] | Nigeria | Cross—sectional survey | 127 participants | 28.1 | 46% | 54% | The results revealed that 87.5% of the 127 respondents had good COVID-19 knowledge | The result on attitude showed that most of the respondents (93.6%) had a positive attitude pertaining to COVID-19 prevention | N/A | Majority of the healthcare workers demonstrated good knowledge, attitude and willingness towards COVID-19 preventive measure |
| Okoro et al and 2020 [42] | Nigeria | An interventional study with a pre- and post-test assessment | (Pretest) 141 participants (Posttest) 134 Participants | 39.28 | 78.7% | 21.3% | This study revealed a high overall knowledge about the disease among the participants | In general, study participants had a positive attitude related to COVID-19 | This study found an overall high level of preventive practice towards the disease. | The study revealed a high knowledge level, practices and attitude among correctional officers towards COVID-19 |
| Olum et al and 2020 [9] | Uganda | Cross sectional study | 136 participants | 32 | 64% | 36% | Overall, 69% of the healthcare workers had sufficient COVID-19 knowledge | Few of the healthcare workers (21%) reported a positive attitude with respect to COVID-19 | Overall, 74% had good practices regarding COVID-19 | Over two-thirds of the healthcare workers had sufficient knowledge on the diagnosis, transmission, and prevention of COVID-19 |

*(Continued)*

**Table 1.** (Continued)

| Author and year | Study setting (Country) | Study design | Population (n) | Mean/Age range of participants | Percentage (%) of males | Percentage (%) of females | Knowledge related to COVID-19 | Attitude/Perception towards COVID-19 | Practice towards COVID-19 | Relevant findings |
|---|---|---|---|---|---|---|---|---|---|---|
| Olum et al and 2020 [48] | Uganda | Cross-sectional study | 741 participants | 24 | 63% | 37% | Overall, 91% of the study participants had good knowledge | Overall, 74% had a positive attitude towards COVID-19 | Overall, 57% had good practices towards COVID-19 | The medical students had sufficient COVID-19- related knowledge and majority reported willingness to engage in the frontline health care response when required |
| Reuben et al and 2020 [43] | Nigeria | Cross-sectional survey | 589 participants | 18–39 years | 59.6% | 40.4% | Most of the respondents (99.5%) had good COVID-19 knowledge | Most respondents (79.5%) showed positive attitudes towards adherence to the government's infection prevention and control (IPC) measures | Majority of the respondents (82.3%) practiced self-isolation/social distancing, used facemask, and improved personal hygiene | The participants in this study had good knowledge and attitudes regarding COVID-19 |
| Sengeh et al and 2020 [53] | Sierra Leone | Cross-sectional survey | 1253 participants | 18 and 39 years old | 52% | 48% | There was high awareness about the novel coronavirus, with 91% indicating that they had heard of COVID-19 | Seventy-five per cent of the respondents felt they were at moderate or high risk of contracting coronavirus in the next 6 months, although response differed widely across regions | A little over half of the respondents reported that they had taken action to prevent COVID-19 infection | The study showed that while COVID-19 awareness and risk perception was high, most respondents do not know that it is possible to survive COVID-19. |
| Tamire and Legesse and 2020 [35] | Ethiopia | Cross-sectional survey | 526 participants | 32.5 | 46.2% | 53.8% | Most of the health care professionals scored above 87.1% for the knowledge questions | Majority of the health care professionals (74.9%) had good attitude that covid-19 will be controlled successfully | This study found that the practice of prevention methods was not satisfactory | The study result showed that there was a huge knowledge gap on the asymptomatic transmission of the disease |
| Tesfaye et al and 2020 [36] | Ethiopia | Cross-sectional survey | 295 participants | 32.2 | 51.5% | 48.5% | Over half of the participants (53.2%) had adequate COVID-19 knowledge | A large proportion of the participants (89.8%) had a positive attitude towards the value of following the WHO guidelines to reduce the transmission of COVID-19 | Many of the study participants (97.3%) engaged in hand washing, which is one of the WHO recommended preventive measures | The findings showed that study participants had a high level of knowledge on different aspects of COVID-19 |

(*Continued*)

**Table 1.** (Continued)

| Author and year | Study setting (Country) | Study design | Population (n) | Mean/ Age range of participants | Percentage (% of males | Percentage (%) of females | Knowledge related to COVID-19 | Attitude/ Perception towards COVID-19 | Practice towards COVID-19 | Relevant findings |
|---|---|---|---|---|---|---|---|---|---|---|
| Umeizudike et al and 2020 [44] | Nigeria | Cross-sectional study | 102 participants | 25.3 | 54.9% | 45.1% | Half of the students (50%) had adequate COVID-19 knowledge | Most of the students (95.1%) had positive attitudes regarding COVID-19 infection control practices | N/A | The overall knowledge of the students regarding COVID-19 was less than adequate although they had positive attitudes towards COVID-19 infection control practices |

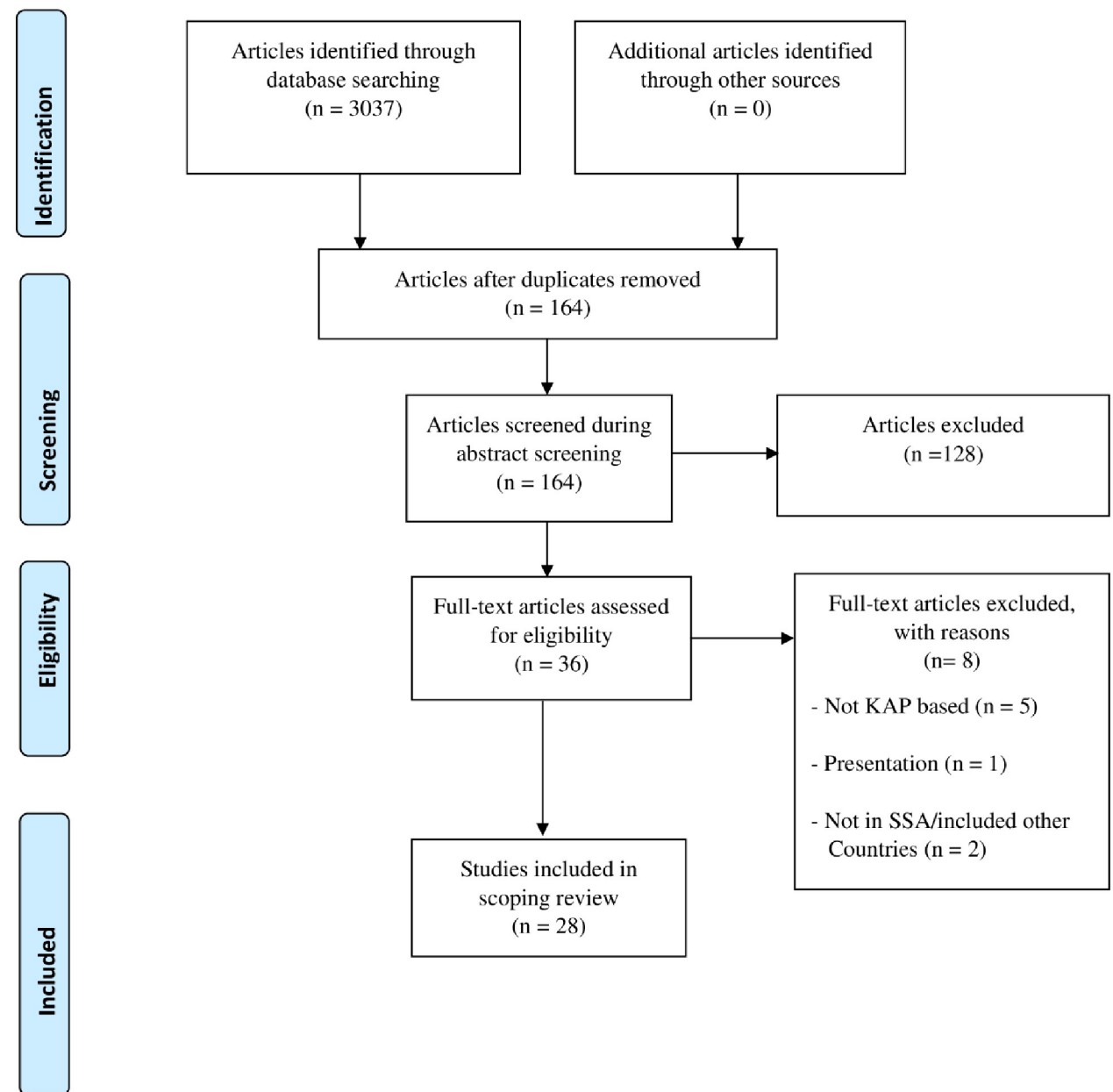

**Fig 1. PRISMA flow-chart of the study selection process.** [Source: Adapted from Moher et al. [28]].

## Characteristics of the included studies

The included studies were all conducted in sub-Saharan Africa and published in 2020. Ten of the studies were conducted in Ethiopia [10, 12, 29–36], eight in Nigeria [37–44], three in Cameroon [45–47], two in Uganda [9, 48], one each in Rwanda [49], Ghana [50], Democratic Republic of Congo [51], Sudan [52], and Sierra Leone [53], (**Fig 2**). A total number of 14,353 study participants were reported in the included studies, with over half (51%) of them being males. Participants' ages ranged from 18 years to 69.5 years old.

Twenty-five out of the 28 included studies were cross-sectional surveys [9, 10, 12, 29–31, 33–39, 41, 43–53], with one each being a web-based survey [32], a descriptive study [40] and

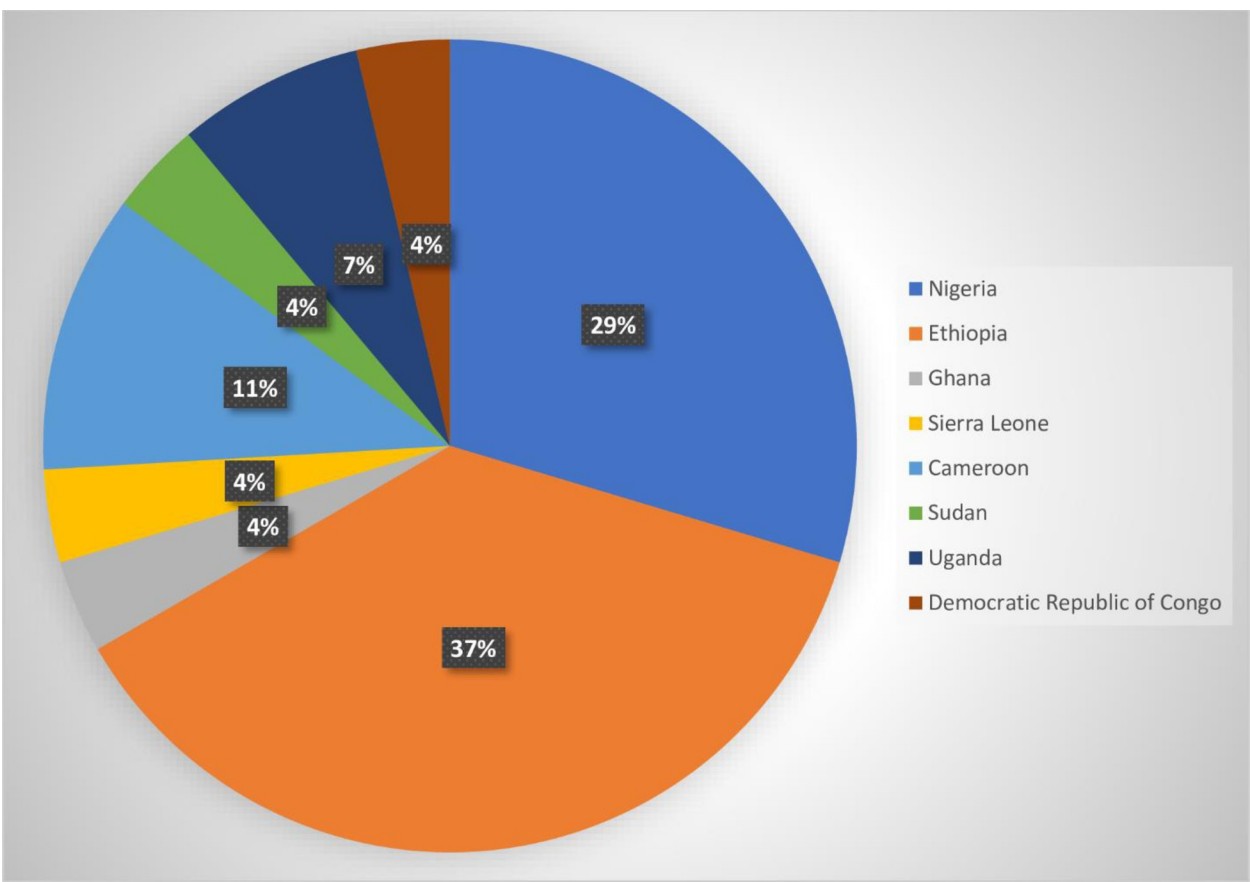

**Fig 2. Distribution of the countries represented in the included studies (N = 28).**

an interventional study with a pre-test / post-test assessment [42]. All included studies showed evidence on the knowledge, attitudes, and practices towards COVID-19 in SSA. **Table 1** illustrates the characteristics of the included studies.

The KAP components of the included studies (**Fig 3**), showed that participants from 24 studies had a good knowledge [9, 10, 29–32, 34–46, 48–50, 52, 53], with only 4 studies reporting a low knowledge of COVID-19 [12, 33, 47, 51]. Participants in fifteen studies had a good attitude/perception [29, 34–37, 41–49, 52], average in five studies [10, 12, 33, 40, 53], and low in four studies [9, 31, 38, 51]. Good preventative practice towards COVID-19 was found in twelve studies [9, 30, 36–38, 42, 43, 45, 46, 49, 50, 52], average in six [10, 12, 29, 39, 48, 53], and low in other six studies [31–33, 35, 47, 51]. Thirteen of the included studies were from the Eastern African region of SSA, with Ethiopia [10, 12, 29–36], Uganda [9, 48] and Rwanda [49] accounting for ten, two and one, respectively, whereas West Africa contributed ten studies, with eight being from Nigeria [37–44], one from Ghana [50], and one from Sierra Leone [53]. A total of four studies were from the Central Africa, with one and three representing the Democratic Republic of Congo [51], and Cameroon [45–47], respectively. Only one study was from North Africa and this is in reference to Sudan [52].

## Quality of evidence from the included studies

Methodological quality assessment was carried out on all the included studies (**S1 Table**) using the Mixed Methods Appraisal Tool (MMAT)-version 2011 [19]. Twenty of the included

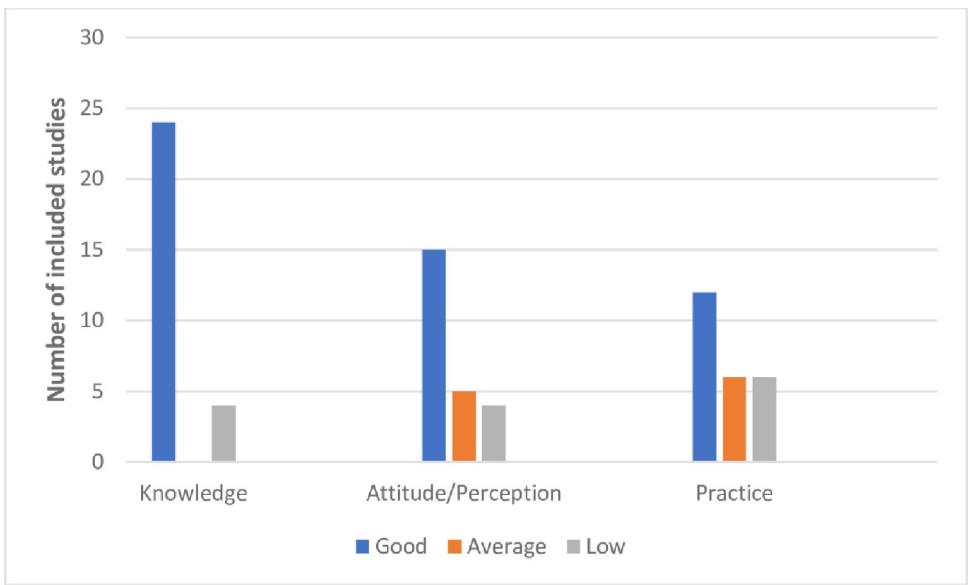

**Fig 3. The KAP components of the included studies.**

studies had a high-quality assessment score of 76–100% [9, 10, 12, 29–39, 42, 45, 47, 48, 50, 53] while the remaining eight studies attained an average quality score of 51–75% [40, 41, 43, 44, 46, 49, 51, 52]. Of the total 28 studies which were quality assessed, none of the studies attained low quality score below 50% thereby showing a minimal risk of bias in the overall evidence.

## Themes from included studies

**Knowledge related to COVID-19.** All included studies reported evidence on knowledge about COVID-19. Majority of the participants in a study conducted in Nigeria showed very good knowledge (99.5%) of COVID-19 [43], this is in contrast to another study carried out in neighbouring Cameroon, which revealed that only 21.9% of the study participants had correct knowledge of COVID-19 [47]. Majority of the participants from the 28 included studies demonstrated high/good knowledge of COVID-19 [9, 10, 29–32, 34–46, 48–50, 52, 53], while less than half of participants from 4 studies had low scores of good knowledge related to COVID-19 [12, 33, 47, 51]. A study in Ethiopia showed that all participants (100%) responded correctly to all preventive knowledge questions [32]. A cross-sectional study conducted in Cameroon revealed that the overall proportion of participants with correct COVID-19 knowledge moved from 9.1% at onset to 41.4% after two months [46]. Most of the participants mentioned that the older population and people with underlying medical conditions were at the highest risk of COVID-19 [12, 30, 36, 44].

Eighteen studies showed that people had good knowledge about the mode of transmission of the COVID-19 virus. They knew that the virus could be transmitted through respiratory droplets of an infected person when they cough, sneeze or speak [10, 12, 29–33, 36, 40–48, 53], whereas participants from three studies had poor knowledge of COVID-19 mode of transmission [35, 37, 51]. Most of the included studies' participants mentioned fatigue, fever, dry cough and myalgia as the major signs and symptoms of COVID-19 [9, 10, 12, 29–37, 39–48, 51, 53]. The participants' major sources of information were television, health professionals, radio, colleagues and social media [9, 10, 29, 30, 32, 34, 36–45, 47, 48, 50–53]. Although certain television channels in Ethiopia (such as Tigrai TV and DW TV), radio station (Fana FM radio

Mekelle branch), and internet service were blocked for a considerable period for unknown reasons since the start of COVID-19. The paucity of COVID-19-related information due to blockage of this media sources could have resulted to the low knowledge scores reported [33].

**Attitudes/perceptions towards COVID-19.** Eleven studies showed that participants had a positive attitude towards COVID-19 [34, 36, 37, 41–45, 47, 48, 52]. In contrast, seven studies revealed the participants' poor attitude towards COVID-19 [9, 29, 31, 33, 35, 38, 49]. A study conducted in Cameroon reported that when attitude concerning COVID 19 health-seeking behaviours was evaluated, majority (73.1%) of the participants believed that they could be infected by health care workers [45]. With regards to people's willingness to carry out a COVID-19 test, a study conducted in Cameroon showed that 72.0% of the participants were willing to undergo a voluntary COVID-19 test, and 47% of these participants preferred obtaining medical care from the house instead of the hospital if tested positive [45]. Participants preferred house medical care due to the fear of being infected in the hospital; they would be well cared for by their families and they would feel comfortable at home [45]. In another study carried out in Ethiopia, 90% of health care providers responded that if they get infected with COVID-19, they will not agree to be isolated in health facilities [34].

Six studies found that majority of participants think they are at risk or may likely get infected with COVID-19 [10, 29, 34, 36, 40, 47], this is contrary to another study where few (36.1%) of the participants were of the perception that they were at risk of infection [31]. Another study conducted in Ethiopia among traditional healers and religious clerics showed that two-thirds of the study participants (66.6%) believed that they would recover if they were infected with COVID-19 [31]. Participants from two studies stated that people could be infected with COVID-19 through packages shipped from infected countries [46, 47].

Four studies conducted in Ethiopia reported that 55.6% [29], 68.9% [33], 69.9% [34] and 74.9% [35] of the respondents, respectively, thought that COVID-19 could be controlled successfully. Similarly, participants from two studies conducted in Sudan and Nigeria, also reported that most of the respondents, 85.1% [52] and 75.9% [42], thought COVID-19 would be successfully controlled. These results contrast with another study where most participants (88%) did not agree that COVID-19 would be defeated and successfully controlled in the DRC [51].

A study by Mandaah et al. [46] revealed that the proportion of participants who reported they would resort to auto-medication in case of an infection with COVID-19 significantly decreased two months after the pandemic began [46]. This positive trend resulted from the implementation of preventive measures by the health stakeholders and government [46]. In a study conducted in Nigeria, more healthcare workers (n = 110, 36.67%) shared unwillingness to treat patients with COVID-19 even if they were well compensated, whereas 80 participants agreed that if they are adequately compensated they will be willing to attend to COVID-19 patient [40].

Nine studies reported that attitude towards COVID-19 preventive measures such as hand hygiene, social distancing, wearing face masks and avoiding crowded areas are essential in controlling the spread of COVID-19 [10, 31, 34, 35, 41, 43, 44, 47, 52]. This is in contrast with some studies where the participants had reported not using masks in crowded areas and when leaving home [29, 52]. Two studies showed that females had more negative attitude towards COVD-19 prevention than males [38, 48]. A study conducted in Ethiopia among traditional healers and religious clerics showed that most respondents believed that COVID-19 was due to God's punishment (n = 356, 86.8%) and traditional healers/religious clerics (n = 278, 67.8%) were better in managing COVID-19 compared to physicians [31]. There is a need for further education to convey the importance of forming a positive attitude to reduce the contraction and transmission of COVID-19.

**Practice towards COVID-19.** Fifteen studies showed that participants had good practices towards COVID-19 [9, 30, 32, 36–38, 42–45, 48–50, 52, 53], while six studies showed poor practice towards COVID-19 [10, 29, 31, 33, 35, 39]. A study conducted in Cameroon reported that two months after the onset of the pandemic, the participants showed improvement in their practices towards COVID-19 and the percentage of participants who obtained face masks and hand sanitizers for their safety was substantially higher compared to the percentage at the onset of the pandemic [46].

Majority of respondents from included studies reported practice towards COVID-19 as wearing a face mask, avoiding crowded areas, observing physical/social distancing, washing hands and using sanitizers [9, 10, 12, 29–33, 35–37, 39, 42–46, 48, 50, 52, 53]. A study conducted in Rwanda among people living with HIV/AIDS showed that most of the participants (90%) had a high practice score [49], and this is in contrast with another study conducted in Ethiopia among traditional healers and religious clerics where only 15.6% of the participants had good practice regarding the prevention and early detection of COVID-19 [31].

## Discussion

The findings of this scoping review reported varied evidence on the knowledge, attitude, perceptions, and preventative practice (KAP) towards novel coronavirus 2019 in SSA. The KAP of the population is very crucial in the control of the virus and significant for policies and intervention efforts.

The included studies revealed that the participants had very good COVID-19- related knowledge except in 4 studies, two conducted in Ethiopia [12, 33], one in Cameroon [47] and another one in the Democratic Republic of Congo [51], where participants had low scores of good knowledge of COVID-19. Two studies conducted in Ethiopia found a high prevalence of poor knowledge of COVID-19, negative attitude, and poor practice among participants [10, 29]. This might be due to the similarities in access to information, socio-demographic characteristics, and awareness [29]. This study found that majority of the participants from the western region had very good knowledge, positive attitude, and good practice towards COVID-19 [37, 38, 41–43, 50, 53], as compared to the eastern region where most participants had low knowledge, negative attitude, and poor practice towards COVID-19 [10, 29, 31, 33]. A possible explanation for the participants' good KAP in the western region could be attributed to the unlimited access to information about COVID-19, disseminated on various media [38], which was different from the eastern region, where most participants had no access to internet and electricity, thereby resulting in limited access to COVID-19 related updates and information [29]. Participants from few studies believe that the governments are not doing enough to prevent and control COVID-19 outbreaks in their countries [9, 33, 34, 40, 43]. Importantly, good knowledge about COVID-19 did not always appear to be a precursor of positive attitude or preventative practices and vice versa. In other words, in some instances, knowledge score was high, but attitude and/or preventative practice was poor [9, 31, 38, 43–45, 49]. A number of pharmacists in Ethiopia had confidence in healthcare facilities' capacity to properly handle potential COVID-19 outbreak in the country [36]. Health authorities in SSA countries need to prioritize the safety of healthcare workers from COVID-19 infection at the healthcare facilities.

This study found that most of the participants from the included studies mentioned dry cough, fever, fatigue and myalgia as the major COVID-19 signs and symptoms. However, few studies showed the wearing of face masks and hand hygiene as a priority, hence these were rarely purchased and few wore face masks when leaving home, despite possessing relatively good knowledge [10, 33, 47, 52]. The practice of physical/social distancing was not adhered to in some studies [10, 12, 31, 51]. In this study, we found that more than half (51.7%) of the

study participants perceived practicing a physical distance as a difficult COVID-19 health protocol [10]. Nonetheless, implementing these preventative measures will likely help to slow down the spread of the virus in the communities.

This study showed a positive trend two months after the onset of the pandemic in the population's knowledge, attitude, and practices towards COVID-19. The positive impact on the positive trend observed was after implementing government and health stakeholders' preventive measures [46].

A study conducted during the lockdown period indicated that the healthcare workers' attitude to work was poor (25.06%), and more females had poor attitude compared to the males [38]. A low percentage of healthcare workers (21%) in Uganda had a positive attitude [9], this is contrary to another study conducted in Nigeria where (93.6%) of the healthcare workers had a positive attitude towards covid-19 prevention [41]. Healthcare workers have an increased risk of infection if their knowledge, attitude, and willingness concerning COVID-19 prevention method is poor.

## Strengths and limitations

To our knowledge, this is the first scoping review to map evidence on the KAP towards the novel coronavirus- 2019 in SSA. A comprehensive search strategy was conducted in this study, which facilitated identifying a considerable number of studies. The scoping review methodology included various study designs and used a systematic approach in identifying relevant studies, charting, and analyzing the selected study outcomes [15]. The results of this scoping review followed the PRISMA guidelines. The review also included a quality assessment of the included primary studies using the MMAT tool to assess risk of bias [19]. However, there is still a possibility that relevant articles were omitted, especially since our search was limited to studies published in English.

## Conclusion

This study showed evidence of high prevalence of knowledge related to COVID-19 in all participants included in the studies. However, there remains a significant gap in the attitude and practice towards COVID-19 in SSA, suggesting that interventions should go beyond just knowledge, but begin to positively affect attitudes and ultimately practices. Therefore, it is important to strengthen health education, information broadcasting, and awareness on the knowledge, attitude, and practice of COVID-19 to slow down this pandemic. There is also a need to make available enough personal protective equipment to health care workers, raise their awareness of infection prevention and control in health facilities in SSA and interventions that improve the community's knowledge, attitude, and practice towards COVID-19 prevention are needed.

## Supporting information

**S1 Appendix. Preferred reporting items for systematic reviews and meta-analyses extension for scoping reviews (PRISMA-ScR) checklist.**
(DOCX)

**S2 Appendix. PubMed search strategy.**
(DOCX)

**S1 Table. Quality assessment of included studies.**
(XLSX)

## Author Contributions

**Conceptualization:** Ugochinyere Ijeoma Nwagbara, Emmanuella Chinonso Osual, Rumbidzai Chireshe, Obasanjo Afolabi Bolarinwa, Balsam Qubais Saeed, Nelisiwe Khuzwayo, Khumbulani W. Hlongwana.

**Data curation:** Ugochinyere Ijeoma Nwagbara, Emmanuella Chinonso Osual, Obasanjo Afolabi Bolarinwa.

**Formal analysis:** Ugochinyere Ijeoma Nwagbara.

**Investigation:** Ugochinyere Ijeoma Nwagbara, Emmanuella Chinonso Osual, Rumbidzai Chireshe, Obasanjo Afolabi Bolarinwa.

**Methodology:** Ugochinyere Ijeoma Nwagbara, Emmanuella Chinonso Osual, Rumbidzai Chireshe, Obasanjo Afolabi Bolarinwa.

**Supervision:** Balsam Qubais Saeed, Nelisiwe Khuzwayo, Khumbulani W. Hlongwana.

**Validation:** Ugochinyere Ijeoma Nwagbara, Emmanuella Chinonso Osual, Rumbidzai Chireshe, Obasanjo Afolabi Bolarinwa, Balsam Qubais Saeed, Nelisiwe Khuzwayo, Khumbulani W. Hlongwana.

**Writing – original draft:** Ugochinyere Ijeoma Nwagbara, Emmanuella Chinonso Osual, Rumbidzai Chireshe, Obasanjo Afolabi Bolarinwa, Khumbulani W. Hlongwana.

**Writing – review & editing:** Ugochinyere Ijeoma Nwagbara, Emmanuella Chinonso Osual, Rumbidzai Chireshe, Obasanjo Afolabi Bolarinwa, Khumbulani W. Hlongwana.

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
