## [Decision Letter · Decision Letter 0]

5 Mar 2021

PONE-D-21-04794

Knowledge, attitude, perception, and preventative practices towards COVID-19 in sub-Saharan Africa: a scoping review

PLOS ONE

Dear Dr. Nwagbara,

Thank you for submitting your manuscript to PLOS ONE. After careful consideration, we feel that it has merit but does not fully meet PLOS ONE’s publication criteria as it currently stands. Therefore, we invite you to submit a revised version of the manuscript that addresses the minor points raised during the review process.

In particular, avoiding duplication of the results description and providing some more details in the Material and Metods section.

We look forward to receiving your revised manuscript.

Kind regards,

Isabelle Chemin, PhD

Academic Editor

PLOS ONE

Journal Requirements:

2. Please ensure that the results of the quality assessment are fully reported in a table the main text showing how each article scored on every item of the scale used.

Reviewers' comments:

Reviewer's Responses to Questions

**Comments to the Author**

1. Is the manuscript technically sound, and do the data support the conclusions?

Reviewer #1: Yes

2. Has the statistical analysis been performed appropriately and rigorously? 

Reviewer #1: Yes

3. Have the authors made all data underlying the findings in their manuscript fully available?

Reviewer #1: Yes

4. Is the manuscript presented in an intelligible fashion and written in standard English?

Reviewer #1: Yes

5. Review Comments to the Author

Reviewer #1: -This is an interesting topic to study due to the nature of the subject matter as it is the subject in the mouth of everyone in the whole world today. The authors have clearly presented their work and in a good professional manner however they may consider these few remarks that could also add value to their great work which they have done.

-In exclusion criteria they may consider to explain the reason for excluding infants, we should not assume that every leader will know it.

-On discussion the authors, could run away from presetting the results again but rather explain their meaning only.

-On results presentation, if they could add some graphs from the results could also add value to the way the results are being presented, for example the graph could show total division of 28 scores on K A P.

-On Results and discussion, the researchers may also consider dividing the sub-Saharan countries in general into regions for regional comparisons to see which region does well and why on KAP.

6. PLOS authors have the option to publish the peer review history of their article (what does this mean?). If published, this will include your full peer review and any attached files.

Reviewer #1: No

---

## [Author Response · Author response to Decision Letter 0]

15 Mar 2021

Author’s response to the reviews 

Title: Knowledge, attitude, perception, and preventative practices towards COVID-19 in sub-Saharan Africa: a scoping review 

Authors:

Ugochinyere I. Nwagbara: 216045259@stu.ukzn.ac.za / ugochinyereijeoma@gmail.com

Emmanuella Chinonso Osuala: 218024179@stuukznac.onmicrosoft.com / osualachinonso@yahoo.com

Rumbidzai Chireshe: 216041773@stu.ukzn.ac.za / rchireshe1@yahoo.ca

Obasanjo Afolabi Bolarinwa: 219098880@stu.ukzn.ac.za / bolarinwaobasanjo@gmail.com

Balsam Qubais Saeed: bsaeed@sharjah.ac.ae

Nelisiwe Khuzwayo: Khuzwayone@ukzn.ac.za

Khumbulani W. Hlongwana: Hlongwanak@ukzn.ac.za

Manuscript PONE-D-21-04794

We are very grateful for the reviews provided by the editor and the reviewers of this manuscript. The comments were very useful. We have revised the manuscript according to the reviewer’s comments. Below is the point-by-point response to the reviewers’ comments. Please find attached a revised version of the manuscript with tracked changes highlighted in ‘red’.

Responses to reviewers’ queries/ comments

Reviewer Comments

This is an interesting topic to study due to the nature of the subject matter as it is the subject in the mouth of everyone in the whole world today. The authors have clearly presented their work and in a good professional manner however they may consider these few remarks that could also add value to their great work which they have done.

Authors’ Response

We appreciate the compliment, thank you.

-In exclusion criteria they may consider to explain the reason for excluding infants, we should not assume that every leader will know it.

Authors’ Response

Thanks, we have included the reason for excluding infants (page 7, lines 151-152).

-On discussion the authors, could run away from presetting the results again but rather explain their meaning only.

Authors’ Response

The authors have rearranged the discussion section and removed the repeated results (page 30-32).

-On results presentation, if they could add some graphs from the results could also add value to the way the results are being presented, for example the graph could show total division of 28 scores on K A P.

Authors’ Response

The authors have added a graph – Fig 3 (page 26, lines 3-5) on the results to demonstrate the KAP components of the included studies and elaborated on it (page 11, lines 264-270).

-On Results and discussion, the researchers may also consider dividing the sub-Saharan countries in general into regions for regional comparisons to see which region does well and why on KAP.

Authors’ Response

Thanks for this great suggestion, the authors have divided the SSA countries into regions on the results (page 11, lines 270-276), and on the discussion (page 31, lines 114-122).

---

## [Decision Letter · Decision Letter 1]

26 Mar 2021

Knowledge, attitude, perception, and preventative practices towards COVID-19 in sub-Saharan Africa: a scoping review

PONE-D-21-04794R1

Dear Dr. Nwagbara,

We’re pleased to inform you that your manuscript has been judged scientifically suitable for publication and will be formally accepted for publication once it meets all outstanding technical requirements.

Kind regards,

Isabelle Chemin, PhD

Academic Editor

PLOS ONE

Additional Editor Comments (optional):

Reviewers' comments:

Reviewer's Responses to Questions

**Comments to the Author**

1. If the authors have adequately addressed your comments raised in a previous round of review and you feel that this manuscript is now acceptable for publication, you may indicate that here to bypass the “Comments to the Author” section, enter your conflict of interest statement in the “Confidential to Editor” section, and submit your "Accept" recommendation.

Reviewer #1: All comments have been addressed

2. Is the manuscript technically sound, and do the data support the conclusions?

Reviewer #1: Yes

3. Has the statistical analysis been performed appropriately and rigorously? 

Reviewer #1: Yes

4. Have the authors made all data underlying the findings in their manuscript fully available?

Reviewer #1: Yes

5. Is the manuscript presented in an intelligible fashion and written in standard English?

Reviewer #1: Yes

6. Review Comments to the Author

Reviewer #1: Let me thank you for taking time and addressing to the suggestions and recommendations that we gave in the previous version, which we believe in one way or another has added value to your great work.

7. PLOS authors have the option to publish the peer review history of their article (what does this mean?). If published, this will include your full peer review and any attached files.

Reviewer #1: **Yes: **George N. Chidimbah Munthali

---

## [Editor Report · Acceptance letter]

6 Apr 2021

PONE-D-21-04794R1 

Knowledge, attitude, perception, and preventative practices towards COVID-19 in sub-Saharan Africa: a scoping review 

Dear Dr. Nwagbara:

I'm pleased to inform you that your manuscript has been deemed suitable for publication in PLOS ONE. Congratulations! Your manuscript is now with our production department. 

Kind regards, 

on behalf of

Mrs Isabelle Chemin 

Academic Editor

PLOS ONE